Manuscript prepared for Hydrol. Earth Syst. Sci.
with version 2014/09/16 7.15 Copernicus papers of the LaTeX class copernicus.cls.
Date: 18 May 2020

# From skill to value: isolating the influence of end-user behaviour on seasonal forecast assessment

Matteo Giuliani[1], Louise Crochemore[2], Ilias Pechlivanidis[2], and Andrea Castelletti[1]

[1]Department of Electronics, Information and Bioengineering, Politecnico di Milano, Piazza L. da Vinci, 32, I-20133 Milano, Italy
[2]Swedish Meteorological and Hydrological Institute (SMHI), Hydrology Research Unit, Folkborgsvagen 17, SE-601 76, Norrkoping, Sweden

**Abstract.** Recent improvements in initialization procedures and representation of large scale hydro-meteorological processes contributed in advancing the accuracy of hydroclimatic forecasts, which are progressively more skillful over seasonal and longer timescales. These forecasts are potentially valuable for informing strategic multisector decisions, including irrigated agriculture, where they

can improve crop choices and irrigation scheduling. In this operational context, the accuracy associated with the forecast system setup does not necessarily yield proportional marginal benefit, as this is also affected by how forecasts are employed by end-users. This paper contributes an integrated framework to quantify the value of hydroclimatic forecasts in terms of potential economic benefit to the end-users, which allows the inference of a relation between gains in forecast skill and gains

in end-user profit. We also explore the sensitivity of this benefit to both forecast system setup and end-user behavioral factors. The approach is demonstrated on the Lake Como system (Italy), a regulated lake operated for flood protection and irrigation supply. Our framework relies on an integrated modeling chain composed of three building blocks: bias-adjusted seasonal meteorological forecasts are used as input to the continentally-calibrated E-HYPE hydrological model; predicted lake inflows

are used for conditioning the daily lake operations; the resulting lake releases feed an agricultural model to estimate the net profit of the farmers in a downstream irrigation district. Results suggest that despite the gain on average conditions is negligible, during intense drought episodes informing the operations of Lake Como based on seasonal hydrological forecasts allows gaining about 15% of the farmers' profit with respect to a baseline solution not informed by any forecast. Moreover,

our analysis suggests that behavioral factors capturing different perceptions of risk and uncertainty significantly impact on the quantification of the benefit to the end-users, where the estimated forecast value is potentially undermined by different levels of end-user risk aversion. Lastly, our results

show an exponential skill-to-value relation where large gains in forecast skills are necessary to generate moderate gains in end-user profit, with the ratio that becomes less demanding during extreme drought events.

## 1 Introduction

Recent advances in initialization procedures (e.g., Ceglar et al., 2018) and representation of large-scale hydro-meteorological processes (e.g., Krysanova et al., 2017) have contributed in greatly advancing the accuracy of hydroclimatic services. State-of-the-art meteorological and hydrological forecast products are increasingly skillful over seasonal and longer time scales and thus become valuable assets for informing strategic decisions contributing to flood protection (e.g., Coughlan de Perez et al., 2017; Neumann et al., 2018), drought management (e.g., Crochemore et al., 2017; Turco et al., 2017), or hydropower production (e.g., Block, 2011; Boucher and Ramos, 2018). Irrigated agriculture is one of the sectors expected to benefit the most from hydroclimatic services to better inform crop choices and irrigation scheduling decisions (e.g., Li et al., 2017; Guimarães Nobre et al., 2019), which strongly depend on the expected hydro-meteorological conditions.

In such operational contexts, forecast accuracy is key to communicate along with hydroclimatic services (Contreras et al., 2020). Accuracy depends on the forecast system setup, which introduces uncertainties that depend on initial hydro-climatic conditions on the forecast date, scenarios of predicted meteorological conditions (e.g., climate model outputs), and sometimes the adopted impact model (Pechlivanidis et al., 2020). At seasonal time scales, probabilistic forecasts are often used to convey these uncertainties, potentially adding value for decision making (see Georgakakos and Graham, 2008; Cloke and Pappenberger, 2009, and references therein).

The idea of moving from forecast accuracy to value has been explored in a few recent studies that quantify the value generated by informing water system operations with perfect or synthetic forecasts (e.g., Turner et al., 2017; Denaro et al., 2017), or a pre-specified real forecast product (e.g., Anghileri et al., 2016; Nayak et al., 2018), in terms of increased system reliability. Only a few studies (e.g., Li et al., 2017; Delorit and Block, 2019) assess the economic value of existing hydroclimatic services in informing the solution of planning problems, which require making single decisions (e.g., selection of crop to cultivate) without considering how they influence analogous decisions in the future.

In this paper, we introduce an integrated evaluation framework that allows the quantification of the value of hydroclimatic services by extending traditional forecast quality assessment methods with estimates of the potential economic benefit of the forecasts in informing operational decisions. The approach is demonstrated on the Lake Como system (Italy), a regulated lake primarily operated for flood control and irrigation supply. Here, our framework supports the inference of a relation between gains in forecast skill and in end-user (farmers) profit over both average as well as extreme drought conditions. The proposed framework relies on a modeling chain composed of three building blocks:

(1) bias-adjusted seasonal meteorological forecasts are used as input to a European-wide hydrological model; (2) predicted lake inflows are then used for conditioning the daily lake operations; (3) the resulting lake releases finally feed a crop growth model to estimate the forecast value in terms of gain in net profit for the farmers in the downstream irrigation district. This combination of a state-of-the-art hydroclimatic service with a detailed model of the Lake Como basin makes our findings particularly valuable for the selected case study area, which is located in the region with the highest share of irrigated areas in Europe (Eurostat, 2019).

In this context, we used our framework to isolate the part of the hydrological modelling chain mostly contributing to the estimated forecast value, as well as to assess the sensitivity of the results on different end-user interpretations of the probabilistic forecast information. Forecast value is filtered by the way end-users make use of the provided information, and there is growing evidence that higher forecast accuracy does not necessarily imply better decisions because of the challenges associated to the human interpretation of forecasts as well as to the communication of probabilistic information (Ramos et al., 2010, 2013; Crochemore et al., 2016). The personal interpretation of uncertainty is indeed a subjective process affected by multiple factors, including the way outcomes are framed, the severity of the event being forecasted, and the personal behavioral attitude of the end-users (Gigerenzer et al., 2005; Joslyn et al., 2009). Individual behaviours and risk perceptions therefore play a key role in influencing the end-user assessment of probabilistic seasonal forecast value (Kirchhoff et al., 2013). However, this point has been so far investigated mostly via serious games, interviews, or direct interactions with decision makers, while our work aims at providing a quantitative analysis of this challenge by simulating how different behavioral attitudes (modeled by specific forecast quantiles capturing increasing levels of drought risk aversion) influence the interpretation of the forecast ensemble and ultimately impact on operational decisions and resulting performance.

The paper is organized as follows: in the next section we introduce the Lake Como study site, while Section 3 describes the proposed evaluation framework. Results and discussion are reported in Section 4, while conclusions and final remarks are presented in the last section.

## 2   Study site

Located in the Italian Alps, the Lake Como basin (Figure 1) is a highly controlled water system, including a large regulated lake (active capacity 247 Mm$^3$) serving a wide irrigation-fed cultivated area (1,320 km$^2$), where maize is the most widely grown and productive crop (52% of the area and 1.5 Mton/year). The hydro-meteorological regime is typical of sub-alpine regions, characterized by dry periods in winter and summer, and peaks in late spring and autumn fed by snowmelt and rainfall, respectively. Snowmelt during May-July is the most important contribution to the accumulation of the seasonal storage, which is then used for irrigation supply in the summer during the peak demand

period. The latter often exceeds the natural water availability and makes the role of the lake operation paramount for the system.

The regulation of the lake has been actively studied since the 1980s (e.g., Guariso et al., 1984, 1986) and is driven by two primary competing objectives: water supply, mainly for irrigation, and flood control in the city of Como, which sits at the lowest elevation on the lake shoreline and hence is exposed to flood risk. The agricultural districts downstream prefer to store snowmelt in the lake to satisfy the peak summer water demands, when the natural inflow is insufficient to meet irrigation

requirements. Yet, storing such water increases the lake level and, consequently, the flood risk. Additional interests are related to navigation, fishing, tourism, and ecosystems, that further challenge the existing water management strategies and motivate the search for more efficient solutions relying on hydroclimatic services. On the basis of previous works (e.g., Castelletti et al., 2010; Giuliani and Castelletti, 2016; Giuliani et al., 2016a; Denaro et al., 2017), these two objectives (both to be

minimized) can be formulated as follows:

- Flood control ($J^F$): the average annual number of flooding days in the simulation horizon, defined as days when the lake level is higher than the flooding threshold of 1.24 m;

- Water supply deficit ($J^D$): the daily average quadratic water deficit between the lake release and the daily water demand of the downstream system, subject to the minimum environmental

flow constraint to ensure adequate environmental conditions in the Adda River. The water demand is given by the sum of the water rights of different users and does not vary across years. This quadratic formulation (Hashimoto et al., 1982) generates hedging strategies that minimize large deficits that would generate crop failures, while accepting small, distributed deficits that can be tolerated by most cultivated crops. Notably, the computation of the water

supply deficit includes a time-varying parameter that penalizes more the deficit experienced after germination to the beginning of phenological maturity, with these crop stages determined by the agricultural district model.

## 3   Evaluation framework

The overall workflow of our evaluation framework relies on an integrated modeling chain composed

of the three building blocks illustrated in Figure 2: (i) the E-HYPE hydrological model produces seasonal forecasts of the Lake Como inflows driven by ECMWF System 4; (ii) the Lake Como operational model designs the optimal lake regulation including the inflow forecasts as additional input in the operating policy that determines the water released by the dam; (iii) the agricultural district model estimates the profit of the farmers in the Muzza district, which is the largest among

the irrigation districts served solely by the Adda River (about 700 km$^2$) as well as the one with the largest water concession (2370 Mm$^3$/yr). A detailed description of each component of the evaluation framework is provided in the next subsections.

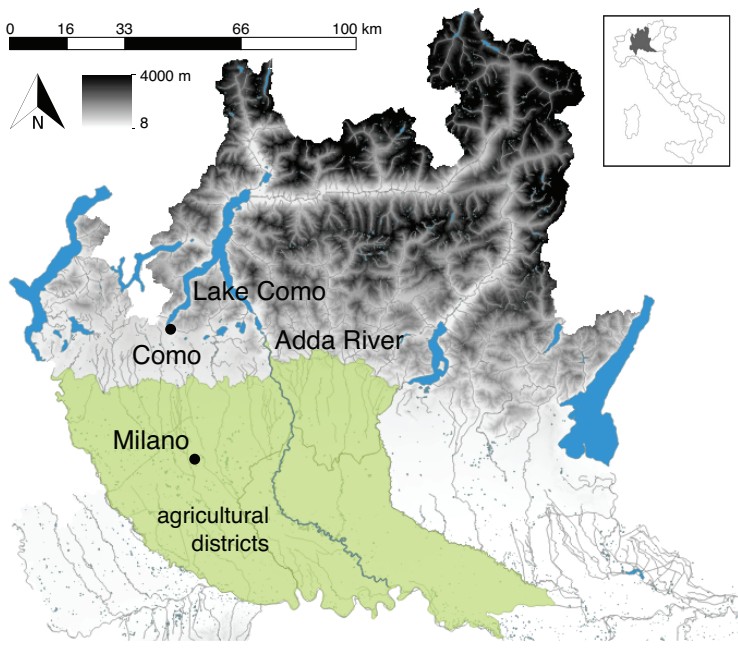

**Figure 1.** Map of the Lake Como basin. The map was generated via Q-GIS using layers from the Geoportal of Regione Lombardia (www.geoportale.regione.lombardia.it).

### 3.1 E-HYPE hydrological model

The European setup of the HYPE hydrological model (E-HYPE; Hundecha et al. (2016)) was used to generate dynamical seasonal streamflow forecasts (Pechlivanidis et al., 2020). E-HYPE is a process-based model that reproduces streamflow and water balance over the entire European continent. Its parameters were calibrated based on a set of 115 catchments representing the diversity of land-use and soil characteristics, as well as human impacts, and over the 1980–1999 period. The model was validated in about 550 catchments for which streamflow observations are available (see details in Hundecha et al., 2016). Here, precipitation and temperature data from the WFDEI reanalysis (Weedon et al., 2014) were used as reference and streamflow simulations were generated by forcing the E-HYPE model with WFDEI meteorological inputs. In the Lake Como basin, E-HYPE exhibits good overall performance in simulating yearly streamflow, though a distinct bias can be seen (Figure 3a). E-HYPE achieves an average yearly root-mean squared error (RMSE) of 748 Mm$^3$/year in the Lake Como basin. This yearly performance hides an underestimation of winter flows, and an overestimation of summer flows at the monthly time step (Figure 3b), which is potentially due to an inaccurate representation of snowmelt dynamics in E-HYPE along with the alterations of the natural hydrologic processes introduced by the operations of the Alpine hydropower reservoirs in the upstream part of the basin. Despite these biases, Crochemore et al. (2020) showed that E-HYPE seasonal forecasts can yield as skilful information as a local model when looking at anomalies relative to model long-

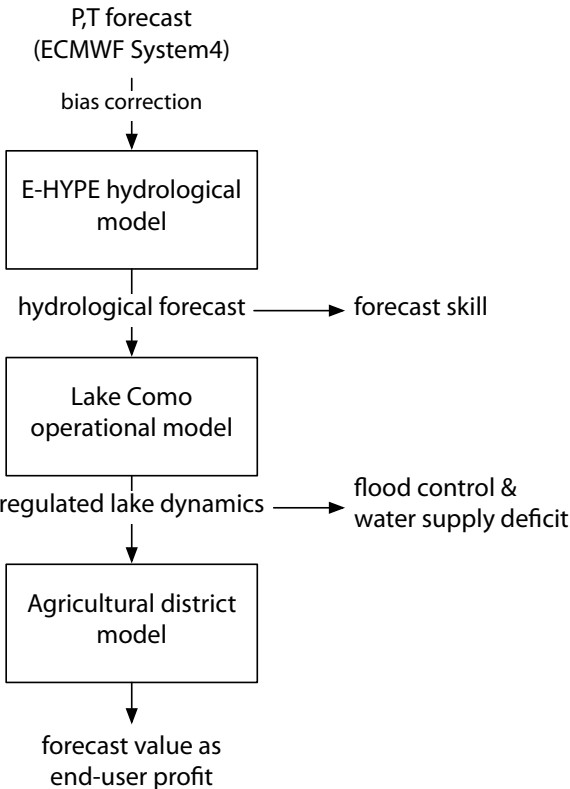

**Figure 2.** Overview of the integrated modeling chain used in the evaluation framework.

term means as done in this work, where the Lake Como operations are optimized using E-HYPE seasonal forecast anomalies.

### 3.2 Operational model of the lake

As mentioned in the previous section, Lake Como is primarily operated looking at two competing
objectives, namely water supply and flood control in the city of Como. The operational model of the lake is focused on reproducing the controlled dynamics of the lake, which is described by a mass balance equation assuming a modeling and decision-making time-step of 24 hours, i.e.

$$s_{t+1} = s_t + q_{t+1} - r_{t+1} \tag{1}$$

where $s_t$ is the lake storage [m$^3$], while $q_{t+1}$ and $r_{t+1}$ are the net inflow (i.e., inflow minus
evaporation losses) and the outflow volumes in the time interval $[t, t+1)$, respectively. The release volume $r_{t+1}$ is determined by a nonlinear, stochastic function that depends on the release decision $u_t$ (Soncini-Sessa et al., 2007). This function allows representing the effect of the uncertain inflows

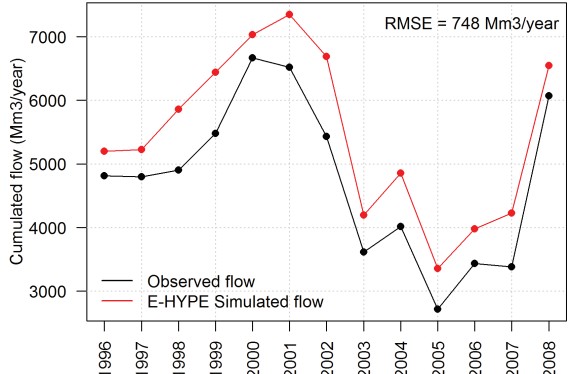

(a) Annual mean cumulated flow

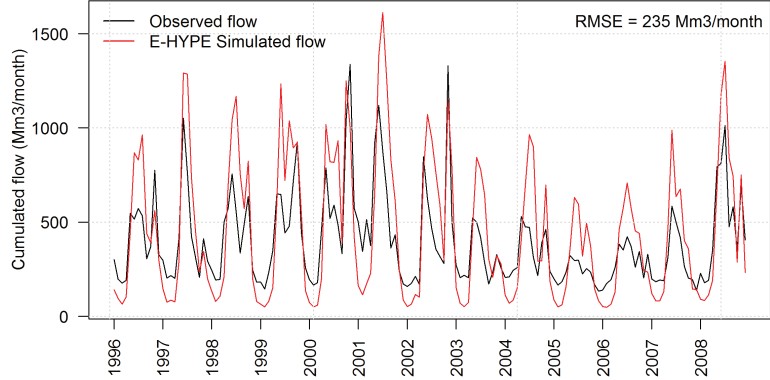

(b) Monthly mean cumulated flow

**Figure 3.** Annual mean accumulated flow (panel a) and monthly mean accumulated flow from observations and E-HYPE simulations from 1996 to 2008 (panel b).

between the time $t$ (at which the decision is taken) and the time $t+1$ (at which the release is completed). The actual release might not be equal to the decision due to existing legal and physical

constraints on the reservoir level and release, including spills when the reservoir level exceeds the maximum capacity.

The lake operations is determined by a closed-loop operating policy $p$ that computes the release decision $u_t$ at each time step $t$ as a function of the day of the year $d_t$, the lake level $h_t$ and the inflow forecast $\hat{q}_{t+\tau}$ over the lead time $\tau$. The Pareto optimal operating policies are computed by solving a

165 multi-objective optimal control problem (Castelletti et al., 2008) formulated as follows:

$$p^* = \arg\min_p \mathbf{J}(p) = |J^F, J^D| \tag{2}$$

Note that the resolution of this problem does not yield a unique optimal solution but a set of optimal solutions exploring different tradeoffs between flood control and irrigation supply. A solution

is defined as Pareto optimal (or nondominated) if no other solution gives a better value for one ob-
jective without degrading the performance in at least one other objective. The image in the objective
space of the Pareto-optimal solutions is the Pareto front. To evaluate the quality of the Pareto front
we used the hypervolume indicator $HV$, which allows set-to-set evaluations by measuring both the
convergence of the Pareto front under examination $\mathcal{F}$ to the optimal one $\mathcal{F}^*$ as well as the repre-
sentation of the full extent of tradeoffs in the objective space (Zitzler et al., 2003). Specifically, this
metric measures the volume of objective space dominated by the considered set of solutions as the
hypervolume ratio between $\mathcal{F}$ and $\mathcal{F}^*$.

### 3.3 Agricultural district model

The agricultural district model simulates the dynamic processes in the Muzza irrigation district. The
model is composed of three distinct modules devoted to specific tasks: (i) a distributed-parameter
water balance module that simulates water sources, conveyance, distribution, and soil-crop water
balance (Facchi et al., 2004); (ii) a heat unit module that computes the sequence of growth stages
as a function of the temperature (Neitsch et al., 2011); (iii) a crop yield module that estimates the
optimal and actual yields, accounting for the effects of stresses due to insufficient water supply that
may have occurred during the agricultural season (Steduto et al., 2009). The water balance module
partitions the irrigation district with a regular mesh of cells with a side length of 250 m, which allows
the representation of the space variability of crops, soil types, meteorological inputs, and irrigation
distribution. Further details about the different model components are provided in Giuliani et al.
(2016c) and Li et al. (2017). In this work we are however not exploring any farmers' decision and
the agricultural district model is therefore not informed by the seasonal forecasts, while the value
of weather and climate services in informing cropping pattern decisions is investigated in Li et al.
(2017).

### 3.4 Data and Experimental Settings

The assessment of the forecast operational value is performed over the time period from January
1, 1996 to December 31, 2008. This period was selected because it shows good variability in the
local hydrological conditions including some intense droughts events that negatively impacted the
agricultural production of the system.

For the purpose of this study, we consider two ensemble streamflow forecasts produced by E-
HYPE. The first one is named *ESP* (Ensemble Streamflow Prediction; Day (1985)) and is generated
by forcing E-HYPE with WFDEI historical scenarios of precipitation and temperature that corre-
spond to the time period of the forecast. The second one is named *SYS4* and uses dynamical precip-
itation and temperature forecasts from the European Centre for Medium-range Weather Forecasts
(Molteni et al., 2011) as input to the E-HYPE model. These forecast inputs are bias adjusted against
the WFDEI reference with the Distribution-Based Scaling method (Yang et al., 2010) prior to run-

**Table 1.** Benchmarking matrix to isolate the sources of forecast value; baseline is observed climatology, ESP E-HYPE Ensemble Streamflow Prediction, SYS4 E-HYPE driven by dynamical precipitation and temperature forecasts, SYS4* replaces the ensemble mean used in SYS4 with different statistics capturing increasing levels of drought risk aversion.

|  | **ESP** | **SYS4** | **SYS4*** |
|---|---|---|---|
| **baseline** | hydrological model + initial conditions | hydrological model + initial conditions + P,T forecast | hydrological model + initial conditions + P,T forecast |
| **ESP** |  | P,T forecast | P,T forecast + behavioral factors |
| **SYS4** |  |  | behavioral factors |

ning the hydrological model. Both ESP and SYS4 forecasts are delivered once a month in the form of a 15-member ensemble with a 7-month lead time. The ensemble means of both ESP and SYS4 are then accumulated over a lead-time of 51 days. This time frame was demonstrated by Denaro et al. (2017) to be the most valuable among different lead times from 1 week to 2 months for improving Lake Como operations. In addition to considering the ensemble means, we investigate the sensitivity of the overall assessment framework with respect to end-user behavioral factors. Specifically, we replace the ensemble mean with the $25^{th}$ and $10^{th}$ percentiles as well as with the ensemble minimum, which capture increasing levels of drought risk aversion. Lastly, the operational value of these two forecast systems is benchmarked against a set of baseline solutions that rely on the local observed climatology and two sets of upper bound solutions using perfect forecasts corresponding to either E-HYPE simulations forced with meteorological observations or the observed lake inflows.

The comparative analysis of results obtained using different forecast products allows isolating the sources of forecast value as illustrated in Table 1. The sources of forecast value include the initial hydrologic conditions, the hydrologic model, the predictions of precipitation and temperature, and the behavioral factors (i.e., the different percentiles of the forecast ensemble considered). In this matrix, each cell identifies the specific forecasting component that is responsible for the differences in farmers' profit using the forecast system indicated on the columns with respect to the benchmark indicated on the rows.

To optimize the operating policy (see eq. 2), we used the evolutionary multi-objective direct policy search method (Giuliani et al., 2016b), a Reinforcement Learning approach that combines direct policy search, nonlinear approximating networks, and multi-objective evolutionary algorithms. The policies are defined as Gaussian radial basis functions (Busoniu et al., 2011) and the policy parameters are optimized using the self-adaptive Borg MOEA (Hadka and Reed, 2013), a combination that has been demonstrated to be effective in solving these types of multi-objective policy design problems featuring the possibility of enlarging the information used for conditioning operational decisions (Giuliani et al., 2015; Zatarain-Salazar et al., 2016; Giuliani et al., 2018). Each optimization

was run for 2 million function evaluations over the simulation horizon 1996-2008. To improve so-
lution diversity and avoid dependence on randomness, the solution set from each formulation is the
result of 20 random optimization trials. The final set of Pareto optimal policies for each experiment
is defined as the set of non-dominated solutions from the results of all the optimization trials. In total,
the analysis comprises 320 million simulations that required approximately 42,670 computing hours

on an Intel Xeon E5-2660 2.20 GHz with 32 processing cores and 96 GB Ram. These high com-
putational requirements explain the use of the water supply deficit as objective in the policy design
rather than the farmers profit, as the latter would require including the simulation of the agricultural
model within the EMODPS optimization substantially increasing the overall computation cost.

## 4    Results and Discussion

### 4.1    Forecast value for irrigated agriculture

Following the proposed evaluation framework (Figure 2), the operational value of alternative forecast
systems can firstly be assessed in terms of improvement in the overall set of Pareto optimal solutions
produced by the use of forecast information using the hypervolume indicator. Then, the simulation
of the agricultural district model will provide a more tangible measure of the forecast operational

value by converting the water supply deficit $J^D$ into monetary values of farmers' profit.

The performance of different sets of solutions obtained by solving the Problem in eq. (2) is shown
in Figure 4a, where each circle represents a different operating policy of Lake Como. The two axes
of the figure represent the two operating objectives (to be minimized) and the arrows indicate the
direction of increasing preference, with the best solution located in the bottom-left corner of the

figure. The comparison of the different Pareto-optimal sets shows large differences in performance
that determine a clear ranking of the generated solutions. Not surprisingly, the use of perfect fore-
casts, either in the form of local observations (black circles) or of E-HYPE simulation (blue circles),
allow designing (ideal) policies that largely outperform the other solutions. The policies using ESP
and SYS4 forecasts are also superior to the baseline solutions, particularly in terms of water supply

deficit values. The considered 51-days lead time is indeed too long to provide valuable information
to control the fast flood dynamics, which is on the order of few days and would therefore require
much shorter lead times. However, the downward shift of the Pareto fronts indirectly influences the
performance in flood control as the new sets of operating policies using forecast information al-
low identifying better compromise alternatives. The numerical quantification of the improvements

in terms of both objectives is provided by the values of hypervolume indicator reported in Table
2, which estimate the ESP and SYS4 forecast values being equal to 6% and 16% of the system
performance, respectively.

To better understand the contribution of the different forecast information to the Lake Como op-
erations, we analyze the dynamic behavior of the system under operating policies that use distinct

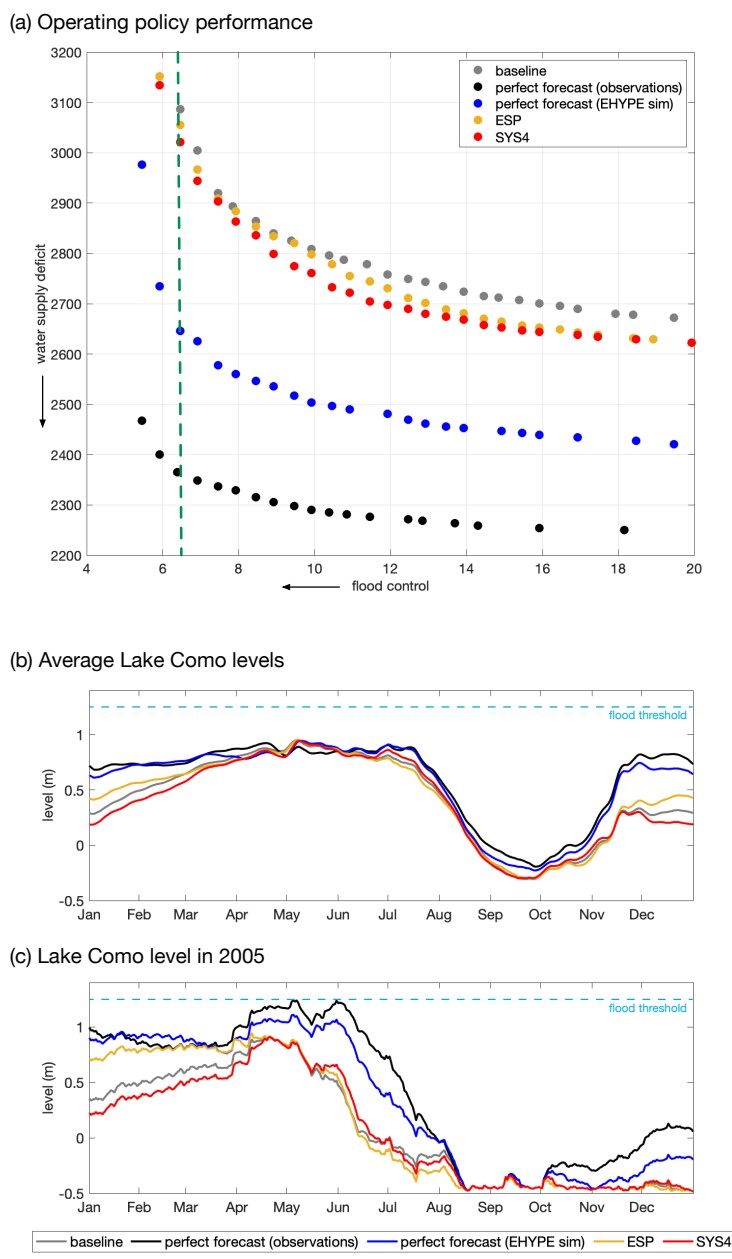

**Figure 4.** Performance obtained by different Lake Como operating policies (panel a) informed with ESP and SYS4 forecasts, along with the upper bound of the system performance (perfect inflow forecasts from observations or E-HYPE simulation) and the baseline operating policies based on observed climatology. The green dashed line marks the performance of the historical lake regulation in terms of flood control. Analysis of average Lake Como levels (measured with respect to the Malgrate reference level at 197.37 m.a.s.l.) under different operating policies (panel b) and during the extreme drought recorded in 2005 (panel c).

**Table 2.** Value of ESP, SYS4, and perfect forecasts in terms of Hypervolume Indicator ($HV$).

| Policies | $HV$ | $\Delta HV$ | relative $\Delta HV$ |
|---|---|---|---|
| Baseline | 0.32 | - | - |
| ESP | 0.34 | 0.02 | 6% |
| SYS4 | 0.37 | 0.05 | 16% |
| Perfect forecast (EHYPE sim) | 0.67 | 0.35 | 109% |
| Perfect forecast (observations) | 1.00 | 0.68 | 212% |

information. This analysis focuses on the solutions located along the green dashed line in Figure 4a, which marks the performance of the historical lake regulation in terms of flood control. The rationale of this choice is to look at solutions that reduce the water supply deficit $J^D$ without degrading the performance in $J^F$. The historical regulation cannot be used as a reference since it also includes additional objectives not accounted for in our model (e.g., navigation, fishing, tourism, ecosystem).

All the simulated trajectories of the Lake Como level under each considered policy show a clear annual pattern, with the highest levels observed in late spring due to the snowmelt contribution (Figure 4b). In this period, maximizing the storage while avoiding floods is crucial to support the summer drawdown cycle driven by high irrigation demands. The policies conditioned on perfect forecast (black and blue lines) are able to maintain the highest level and to delay the drawdown. Conversely,

the baseline solution (gray line), which has no information about future inflows, reaches the highest level at the beginning of May and, subsequently, the level is maintained about 10 cm below the perfect forecast trajectory to have space for buffering potential floods. A similar trajectory is followed by the policy informed by ESP and SYS4 forecasts (orange and red lines, respectively), which are on average almost overlapped until the third week of June, while they look more separated during

the drawdown period with the SYS4 that is able to keep a high level also in July. In addition to the average levels, it is interesting to investigate how the different solutions operate the lake during the extreme drought recorded in 2005 (Figure 4c). The low inflows experienced during this drought event produced an early drawdown of the lake level starting at the beginning of June, when the downstream water demand is at its maximum, with the levels reaching the lower limit of -0.50 m

around middle August. This extreme event confirms and emphasizes the differences observed on the average lake levels; the policies conditioned on perfect forecast maintain the highest level from April to mid-August thus delaying the drawdown. ESP and SYS4 forecasts, although less efficient than the perfect forecast solutions, are able to keep higher lake levels than the baseline solution from mid-May to the beginning of July, thus reducing the water supply deficit. ESP and SYS4 solutions

then reach lower levels than the baseline in the second half of the 2005 summer. This strategy can be considered as an extreme drought mitigation measure triggered by the extreme drought conditions

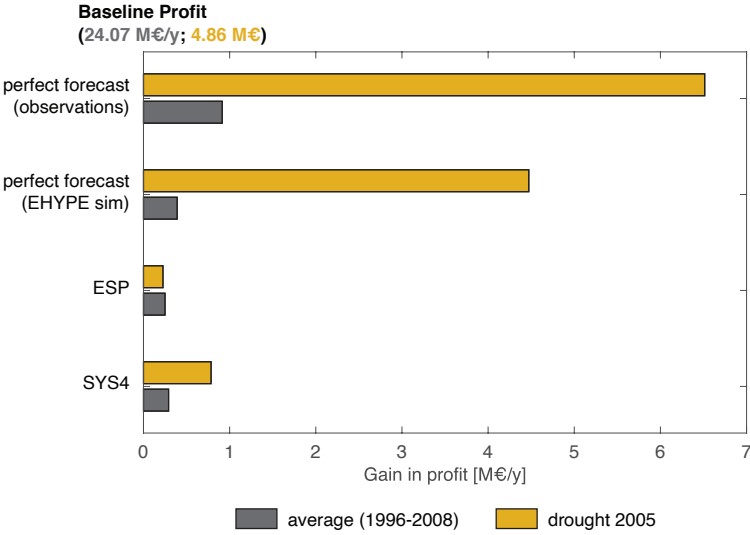

**Figure 5.** Comparison of gains in farmers' profit with respect to the baseline solution under different Lake Como operating policies informed with ESP and SYS4 forecasts, along with the upper bound of the system performance (perfect inflow forecasts from observations or E-HYPE simulation).

predicted for August in order to support a more reliable irrigation supply than under the baseline operations by sacrificing few extra centimeters of lake level.

This analysis can be translated into economic terms via simulation of the Agricultural district model, which estimates the crop production and the associated net profit (i.e., gross revenue minus production costs, also accounting for the EU Common Agricultural Policy subsides (Gandolfi et al., 2014)) for the farmers in the Muzza irrigation district served by the Lake Como releases under different operating policies. Figure 5 shows the same ranking of solutions obtained in the space of the operating objective (Figure 4a), with the use of forecast information that allows gaining, on average, from 1% (ESP forecast) to 3.8% (perfect forecasts from observations) of annual farmers' profit (i.e., from 300,000 €/year to 900,000 €/year) in comparison to the 24.07 million €/year attained by the baseline solution. Interestingly, these values are much larger when evaluated over the 2005 drought, when the baseline annual profit is only 20% of the 1996-2008 average value. In this case, the perfect forecasts generate a profit that is 134% (observations) and 92% (E-HYPE simulation) higher than the baseline; the value of ESP and SYS4 also grows, producing a 5% and 16% increase in farmers' profit, respectively. These results suggest a large potential for using E-HYPE forecast in the management of extreme droughts.

### 4.2 Impact of forecast system setup and behavioral factors on forecast value

Following the benchmarking analysis in Table 1, we investigate the isolated sources of forecast value by assessing the sensitivity of the farmers' profit on both forecast system set up and end-user behavioral factors. For the former aspect, we compare our baseline solution against the operating policies informed by ESP and SYS4 forecasts (using the ensemble means). For the latter, we explore increasing levels of risk aversion in the use of SYS4 forecasts by informing the operating policy with the $25^{th}$ and $10^{th}$ percentiles as well as the minimum of the forecast ensemble.

The results are reported in the comparative matrix in Table 3, which shows again the superiority of ESP and SYS4 over the baseline. Interestingly, the role of predicted precipitation and temperature in drought conditions differs from the average conditions. The use of SYS4 instead of ESP in 2005 generates a 11% gain in farmers' profit, while this difference drops to 0.2% in average conditions. Over the full period, the most important components of the forecast system are the hydrological model and the initial conditions, which together produce more than 1% increase in farmers' profit. Hydrological initial conditions provide the most similar gains between the entire period and the 2005 dry conditions, suggesting that this component is the least sensitive to hydrological conditions. The analysis of the behavioral factors shows that the potential operational value of SYS4 depends on the level of risk aversion used in interpreting the information provided by the forecast ensemble. The average 1.2% increase in farmers' profit with respect to the baseline using the ensemble average grows to 1.35% when the policy is informed by the ensemble minimum, probably because E-HYPE generally overestimates observed inflows (Figure 3a) and predictions of winter low flows are more interesting for managing drought risk. However, results do not demonstrate a linear relationship between forecast value and risk aversion, with the average gain over the baseline being 1.16% when using the $10^{th}$ percentile of the ensemble (which is equal to the gain produced by the ensemble mean) and 0.9% when using the $25^{th}$ percentile of the ensemble.

In addition, our results show that the average contribution to the forecast value of predicted precipitation and temperature (+0.12%) is comparable to the one of the isolated behavioral factors. A solution that uses the ensemble minimum produces a profit 0.14% higher than using the ensemble mean (+0.31% with respect to ESP), whereas the $25^{th}$ percentile of the ensemble generates a 0.31% reduction (-0.14% with respect to ESP). This means that the added value of SYS4 meteorological forecasts can be potentially undermined if end-users are not able to properly extract the most valuable information from the forecast ensemble. However, it should be noted that our results also show that there is not a single best statistic that consistently provides the most valuable information for improving the Lake Como operations. In average conditions, using the ensemble minimum marginally improves the farmers' profit with respect to all the other solutions informed by SYS4 forecasts; conversely, during the 2005 drought, the $10^{th}$ percentile results to be more valuable than the minimum. The use of risk averse statistics in interpreting the forecast ensemble is therefore recommended for water supply operations exposed to drought risk, but more extensive investigations over multiple

**Table 3.** Results of benchmarking analysis to isolate the sources of forecast value. The matrix reports the percentage change in farmers' profit for the forecast systems on the columns with respect to the benchmarks on the rows, estimated as average over the 1996-2008 period and for the 2005 drought (in parenthesis).

|  | ESP | SYS4 - mean | SYS4 - min | SYS4 - p10 | SYS4 - p25 |
|---|---|---|---|---|---|
| **baseline** | 1.04 (4.65) | 1.21 (16.13) | 1.35 (36.26) | 1.16 (40.80) | 0.90 (22.32) |
| **ESP** |  | 0.17 (10.97) | 0.31 (30.20) | 0.12 (34.54) | -0.14 (16.88) |
| **SYS4 - mean** |  |  | 0.14 (17.34) | -0.05 (21.25) | -0.31 (5.33) |
| **SYS4 - min** |  |  |  | -0.19 (3.33) | -0.45 (-10.23) |
| **SYS4 - p10** |  |  |  |  | -0.26 (-13.13) |

extreme events and, possibly, across different case studies is necessary to provide general recommendations.

### 4.3   From forecast skills to end-user value

Lastly, we aim to identify a relation between the increase in forecast skill and the resulting gain in farmer profit from isolated forecast system components. The general assumption is that a gain in

forecast performance should result in a gain in profits; however, the gain in farmer profit might be particular sensitive to having good forecast skill in specific period of the year (see Appendix A for details about the selected time periods for the computation of the forecast skill). Figure 6 shows that the most skillful hydrological forecasts are able to provide the maximum higher conversion rate of skill into end-user value (i.e., farmers' profits), with the overall skill-profit relation well aligned over

an exponential function (i.e., the fitted function attains a $R^2 = 0.965$).

Results also show that the correct assimilation of hydrological conditions on the forecast issue date (i.e., ESP) yields the greatest and only significant gain in skill over the 1996-2008 period (squares). This 10.7% gain in skill obtained by initializing the hydrological model is associated with a 1.04% gain in average farmers' profit. SYS4 forecasts yield a 2% gain in skill which leads to a 0.17% gain

in profit. These results suggest a 10 to 1 relation between skill and profit when the entire period is considered. In this case, the behavioral factors considering low percentiles of the forecast distribution lead to losses in the skill of the (deterministic) information extracted from the forecast ensemble (white squares). These forecasts are associated to small losses and gains in profit that are not systematic and hardly interpretable, suggesting that risk averse behaviors are likely not relevant in average

hydrological conditions.

In the case of 2005 (circles), behavioral factors yield the greatest gains in skill. Focusing on the $10^{th}$ percentile of the forecast distribution yields gains in profit and skill of 21.2% and 40.9%, respectively, whereas focusing on the minimum of the forecast distribution yields gains of 17.7% and 39.3%, respectively. In these cases, the skill to profit relation becomes 2 to 1, while this relation

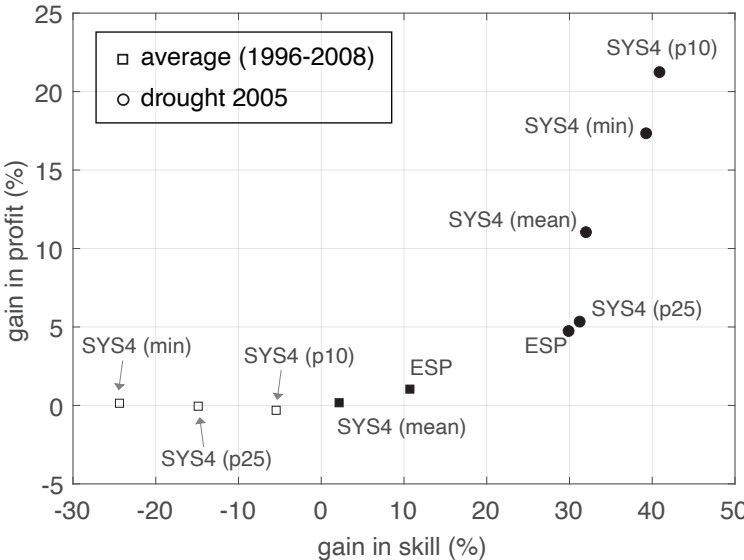

**Figure 6.** Exponential relationship between forecast skill and operational value. Each marker represents either an isolated component of the forecast system or a behavioral factor evaluated over the 1996-2008 period (circles) or the 2005 drought (squares); filled markers identify a positive gain in forecast skill.

decreases into a 3 to 1 for the SYS4 forecast and into a 6 to 1 for the ESP forecast. Overall, these results confirm that improving the skills of seasonal forecast is expected to be particularly valuable to inform the management of extreme events.

### 4.4 Limitations and future research

A limitation in the presented results is the relatively small number of points used to fit the forecast
skill-value relationship. While it would certainly be interesting to repeat the analysis across multiple drought events as well as across different case studies characterized by diverse hydroclimatic regimes, in the context of this work we preferred to perform the analysis using highly detailed models whose associated computational requirements limit the possibility of easily increasing the sample size.
Moreover, it could be interesting to verify if the conclusions drew by Crochemore et al. (2020) hold for the Lake Como basin by comparing the skill and value of E-HYPE forecasts against the ones generated by a fine-tuned local hydrologic model. Extending the economic analysis to other irrigated agricultural systems as well as other sectors (e.g., hydropower, flood protection) is also warranted. Finally, it would be interesting to assess the value of hydroclimatic services under a
projected future climate characterized by more frequent and intense extreme events, which can make forecast information more valuable than under the historical climate.

## 5 Conclusions

In this paper we showcase an integrated evaluation framework to quantify the value of hydroclimatic services in terms of added economic benefit of the forecasts in informing end-user decisions. Moreover, we analyze the isolated sources of forecast value in terms of both forecast system set up and end-user behavioral factors, and we also infer a relation between gains in forecast skill and gains in end-user value. The framework is applied to the operations of Lake Como in the Italian lake district.

Numerical results demonstrate the potential of the E-HYPE hydrological forecast to inform the operations of Lake Como, generating an average 290,000 €/year gain in the net profit of the farmers served by the lake releases (about 1% of the average profit obtained by a baseline solution without forecast information). This gain rises up to 16% (i.e., 800,000 € against a baseline profit equal to 4.9 M€) during the extreme drought experienced in 2005.

The analysis of the isolated sources of the estimated forecast value attributes the largest share of value to the initialization of the hydrological forecasts with conditions relevant to the forecast issue date. For the extreme drought of 2005, the forecast value is instead mostly attributable to the use of precipitation and temperature predictions and to risk averse decisions focused on the lowest part of the forecast ensemble. In addition, our framework shows the need of transitioning from forecast skill assessment to integrated frameworks that include decision models and account for end-user behavioral factors capturing different perception of risk and uncertainty. Investing in advanced training for decision makers and reservoir operators is expected to be crucial for maximizing the uptake of forecast information and their operational value (Crochemore et al., 2016). Conversely, the added value of hydroclimatic services might be undermined if end-users are not able to adequately interpret the uncertainty associated to the forecast ensemble. Lastly, our results suggest an exponential skill-to-value relation where large gains in forecast skills are necessary to generate moderate gains in end-user profit. However, during the 2005 drought, this relationship is less demanding, suggesting that a 10% increase in profit can be obtained with a 30% improvement in forecast skill.

## Appendix A: Analysis of sensitive intra-annual periods for forecast skill

Despite the general assumption that a gain in forecast performance should result in a gain in profits, when relating profits to performance averaged over all months of the year we observed that a loss in skill sometimes resulted in a gain in profit. This suggests that the profit is sensitive to different periods of the year, with the critical intra-annual period that may vary if we focus on the entire study period (1996-2008) or on dry years such as in the example of 2005. A simple sensitivity analysis was thus carried out to identify the months of the year that explain and impact the calculated profits the most. All possible continuous combinations of months were successively tested to compute the forecast skill, which was then related to the estimated profits. When relating profit and skill over the 1996-2008 period, the profit is mostly related to the skill computed over the April to December

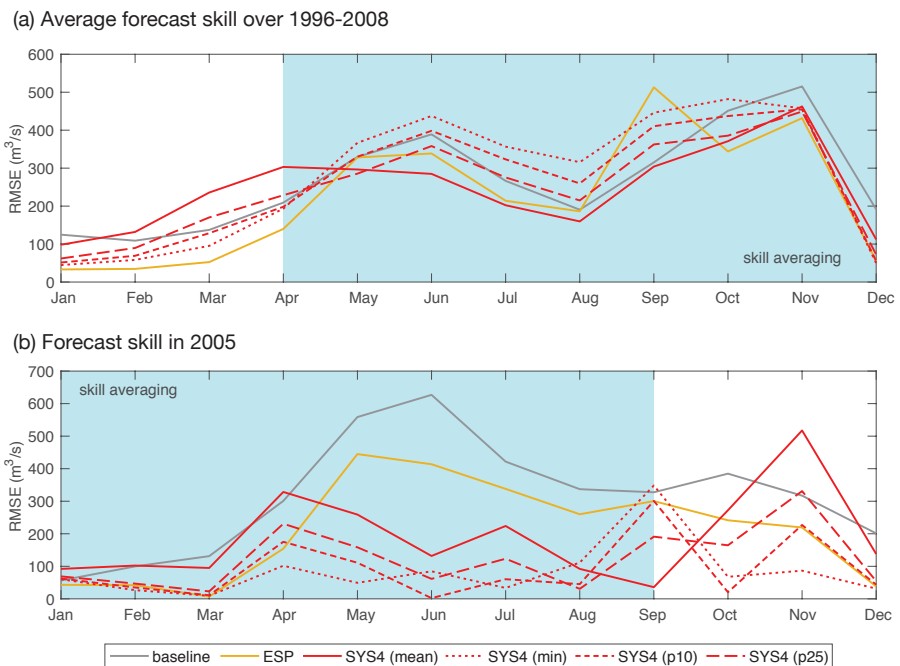

**Figure 7.** Performance of the forecasts investigated in terms of root mean squared error and over (a) 1996-2008 and (b) 2005. The cyan shaded area represents the time of the year when the skill can be related to the profit evaluated over the corresponding period.

period (Figure 7a). When relating profit and skill for 2005, the profit well aligned with the skill averaged from January to September (Figure 7b).

These results are consistent with the strategies adopted in the operation of Lake Como, where the
period from April to September corresponds to the agricultural season. Forecasting and managing that period correctly will always play an important role on the yearly profit. In addition, the fall season also plays an important role for the multipurpose operation of the lake, since intense precipitation events cause generate high risk of flooding. Conversely, in dry years, predicting the filling up of the lake at the end of the winter season is more crucial than predicting winter flooding events, since the
latter have low probability of occurrence in dry conditions. In the considered 2005 drought, the lake operations benefit from skillful forecast also during the period from January to March (Figure 7b).

*Data and code availability*: The seasonal meteorological forecasts SEAS5 from the European Centre for Medium-Range Weather Forecasts are freely accessible from the Copernicus Climate Data Store
(https://cds.climate.copernicus.eu/). The HYPE model code is available from the HYPEweb portal (http://hypeweb.smhi.se/model-water/). Real-time seasonal forecasts obtained through E-HYPE are

openly available on the HYPEweb portal (http://hypeweb.smhi.se/explore-water/forecasts/seasonal-forecasts-europe/). Local observations of lake inflows along with the other meteorological variables used by the agricultural district model were provided by Consorzio dell'Adda (http://www.addaconsorzio.it) and by Agenzia Regionale per la Protezione dell'Ambiente (http://ita.arpalombardia.it). The source code for the Lake Como simulation and EMODPS implementation is available on Github (https://github.com/mxgiuliani00/LakeCom

*Acknowledgements.* The work has been partially funded by the European Commission under the IMPREX project belonging to Horizon 2020 framework programme (grant n. 641811). Funding was also received from the EU Horizon 2020 project S2S4E (Sub-seasonal to seasonal forecasting for the energy sector) under grant agreement No. 776787.

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
