# Peer review of "From skill to value: isolating the influence of end-user behaviour on seasonal forecast assessment"

_Hydrology and Earth System Sciences, 2019_

## Referee Comment (RC1) · Anonymous Referee #1 · 7 Feb 2020

**GENERAL COMMENTS**

The subject of the paper " *From skill to value: isolating the influence of end-user behaviour on seasonal forecast assessment*" is of direct interest to the Journal of Hydrology and Earth System Sciences. Authors introduce and apply a framework in the context of valuing the potential benefit of seasonal forecast in terms of economic end users benefit. Although there are several aspects that need to be further elaborated, this is a step forward in moving from skill to impact (financial) based assessments.

Regarding the different aspects of the HESS journal:

| 1 | Does the paper address relevant scientific questions within the scope of HESS? | YES |
|----|----|----|
| 2 | Does the paper present novel concepts, ideas, tools, or data? | YES |
| 3 | Are substantial conclusions reached? | YES |
| 4 | Are the scientific methods and assumptions valid and clearly outlined? | YES |
| 5 | Are the results sufficient to support the interpretations and conclusions? | YES |
| 6 | Is the description of experiments and calculations sufficiently complete and precise to allow their reproduction by fellow scientists (traceability of results)? | YES |
| 7 | Do the authors give proper credit to related work and clearly indicate their own new/original contribution? | YES |
| 8 | Does the title clearly reflect the contents of the paper? | YES |
| 9 | Does the abstract provide a concise and complete summary? | YES |
| 10 | Is the overall presentation well structured and clear? | YES |
| 11 | Is the language fluent and precise? | YES |
| 12 | Are mathematical formulae, symbols, abbreviations, and units correctly defined and used? | YES |
| 13 | Should any parts of the paper (text, formulae, figures, tables) be clarified, reduced, combined, or eliminated? | NO |
| 14 | Are the number and quality of references appropriate? | YES |
| 15 | Is the amount and quality of supplementary material appropriate? | YES |

**SPECIFIC COMMENTS**

1. One of my concerns is that the examination of the value of forecast is limited to a single lead time (51 days ahead) and the potential benefit of other lead times to the current framework are not examined, or at least discussed.
2. Crop yield modeling is an integral part of the valuing framework. The simulation of crop production is based on water availability and growing degree days controlled by temperature. From the information provided in the manuscript it is not clear whether the heat unit module of agricultural model is also informed by seasonal forecasts.
3. I understand that the present study, beside other components, examines the usefulness/applicability of a continental scale hydrological model (E-HYPE) with known issues in simulating streamflow dynamics due to local scale hydrological features (as referred in L120-

125 – constant positive bias of E-HYPE / failure in seasonal dynamics).The question is whether the use of fine-tuned local scale model would increase the performance of the overall system?

4. Finally, the manuscript would benefit from considering a section summarizing the limitations of the study and ways to overcome these limitations. This could be included in the discussion section.

Considering these, and the fact that the scientific significance and quality are excellent, my suggestion to the editors is to accept after minor revision in the context of my specific and technical comments. I am listing a number of suggestions in the form of technical comments that will improve the presentation of the study.

**TECHNICAL COMMENTS**

L200: The simulation horizon for the policy optimization is 2007-2015 while results are presented for the 1996-2008 period (thereafter). In case this is correct, is there any effect from potentially different operation policies between these two periods (considering also that the 2005 drought is out of the 2007-2015 bound)?

Figure 4: Please consider adding a straight line in panels (b) and (c) indicating flood level.

L240-241: This is not clear in the figure. Please explain.

L249-250: but also less efficient onwards (from July to mid-August).

Figure 5: could consider adding a second panel on the right illustrating the benefit with respect to the baseline.

Table 2: Based on the values in the table, does the optimum profit comes from informing farmers with the minimum values (SYS4-min)?

L279: Please provide more information on the behavioral factors.

---

## Referee Comment (RC2) · Anonymous Referee #2 · 19 Feb 2020

This is a study into the value of inflow forecasts in water release decision making, focusing on the benefits to agricultural profitability. Previous studies have demonstrated how forecasts can be adopted in reservoir operations to marginally improve on a prespecified objective. This paper offers an incremental advance by coupling the reservoir model to an agricultural model, allowing for calculation of profits associated with the updated release schedule. The subject is certainly of general interest to the hydrological community. The paper is well written and the method easy to follow. While the study is mostly sound (I have a few concerns outlined below), a significant contribution to knowledge is missing. One can easily deduce without this study that reduced water supply deficits in a reservoir release model should lead to increased profits for

crop-growers relying on that water. The monetary values provided cannot be offered as a contribution either, since they are not reflective of actual profit gains that would be gleaned by crop-growers (partly because the reservoir operations are stylized for this case study rather than representing real world operations). The approach taken is described as a "novel evaluation framework". It appears to be a forecast product providing information for a reservoir model, the release from which forces an agricultural profit model. One-way coupling of models (which is what I understand this to be) does not qualify as a novel framework in my view. Lastly, because the study is conducted on a single site and using a short time series with only one drought (with much of the analysis drawn from performances during that drought), the conclusions are not generalizable. The authors acknowledge this lack of generalizability in their final sentences, and I think that their suggestions for future research are actually required in this paper to help with the knowledge contribution. While a single case study can be valuable, I cannot see compelling new insights on value of forecasts arising from this analysis to warrant publication. I think the suggested exponential relationship between forecast skill and farmer profit could be a significant contribution if demonstrated and elaborated more carefully through more detailed analysis across a range of possible droughts and with incremental adjustments to the forecast skill. I would be supportive of publication of this paper if the authors can (a) deepen their analysis to generate a more compelling advance on existing knowledge, and (b) address the small number of other concerns listed below.

Specific comments:

The decision to use quadratic water supply deficit as the objective function is not fully justified. If the purpose of the water supply is to meet farmer needs, and if profit is the goal of the farming community, then why not use farmer profit as the objective? This would greatly improve the interpretability of the results, particularly for aim (iii) "the inference of the relation between gains in forecast skill and gains in end-user profit." Currently, the paper lacks discussion on how the discontinuity between the optimization

objective and profitability affects the conclusions drawn. In particular, the squaring of the water deficit objective would normally result in hedging behavior that would reduce overall profit to avoid possible cases of extreme deficit. It's also not clear from this analysis how water deficit affects profit. Does a small deficit necessarily imply loss of crop production, or can farms run at full profitability during periods of small or moderate deficit?

The decision to vary the ensemble selected from mean to 10th and 25th percentiles to capture drought risk aversion requires better justification, too. It would seem more prudent to adjust the objective to represent risk-averse preference (e.g., increasing the exponent applied to the objective, or, if changing the objective function to farmer profit, adding penalties for very significant losses) than to deliberately under-estimate the inflow.

Line 81: please clarify what "heavily man-overworked" means (and why its relevant).

Line 89: do you mean "most Southerly" point on the lake shoreline, or the "near the outflow" of the lake?

Line 257: Why bother with the Pareto analysis if the flood objective effectively becomes a constraint. I don't think the readers of the study need all of the detail of the Pareto analysis if multi-objective optimization is not actually used to generate the key results.

Line 260: the fact that profits are improved through operations is used to support the idea that forecasts can be valuable for managing extreme drought. Presumably the impact is greatest during drought because this is the only time when profit can be compromised (i.e., average flow conditions are unlikely to lead to supply deficits, meaning forecasts are not actually useful except leading up to and during drought). Is this correct? If so, why not focus analysis on droughts and also introduce other drought events to help support and generalize these conclusions?

Line 311: Has this function been fitted across all of the points on Figure 6? Please

justify or comment on the appropriateness of combining the all-years and 2005 results in the same function. The idea of exponential relationship between profit and forecast skill would be a powerful conclusion, but is surely best demonstrated using (a) a model that can adjust forecast skill incrementally allowing generation of many data points, and (b) repeating the analysis across multiple droughts.

---

## Author Comment (AC1) · 31 Mar 2020

The subject of the paper "From skill to value: isolating the influence of end-user behaviour on seasonal forecast assessment" is of direct interest to the Journal of Hydrology and Earth System Sciences. Authors introduce and apply a framework in the context of valuing the potential benefit of seasonal forecast in terms of economic end users benefit. Although there are several aspects that need to be further elaborated, this is a step forward in moving from skill to impact (financial) based assessments.

1. One of my concerns is that the examination of the value of forecast is limited to a single lead time (51 days ahead) and the potential benefit of other lead times to the

current framework are not examined, or at least discussed.

Our analysis focuses on a specific forecast lead time that was identified in a previous work, i.e., Denaro et al. (2017), as the most valuable for improving Lake Como operations. In that work, we did indeed comparatively analyze forecasts over different lead times from 1 week to 60 days. We will better clarify this point in the revised manuscript.

*Denaro, S., D. Anghileri, M. Giuliani, and A. Castelletti (2017), Informing the operations of water reservoirs over multiple temporal scales by direct use of hydro-meteorological data, Advances in Water Resources, 103, 51–63*

2. Crop yield modeling is an integral part of the valuing framework. The simulation of crop production is based on water availability and growing degree days controlled by temperature. From the information provided in the manuscript it is not clear whether the heat unit module of agricultural model is also informed by seasonal forecasts.

The agricultural model is not informed by the forecasts because our analysis investigates the value of forecasts in informing the Lake Como operation that provides the irrigation supply to the agricultural districts considering as water demand the sum of the water rights of the different users, which therefore does not vary across years. Conversely, we are not exploring here decisions by the farmers that could benefit from the seasonal forecasts, but we studied this problem in a previous work (see Li et al., 2017). In the revised manuscript, we will better clarify the decisions we are considering as well as the definition of the irrigation demand.

*Li, Y., M. Giuliani, and A. Castelletti (2017), A coupled human–natural system to assess the operational value of weather and climate services for agriculture, Hydrology and Earth System Sciences, 21, 4693-4709*

3. I understand that the present study, beside other components, examines the usefulness/applicability of a continental scale hydrological model (E-HYPE) with known issues in simulating streamflow dynamics due to local scale hydrological features (as

referred in L120-125 – constant positive bias of E-HYPE / failure in seasonal dynamics).The question is whether the use of fine-tuned local scale model would increase the performance of the overall system?

We agree with the reviewer that a fine-tuned local scale model may in principle increase both the skill and the value of the forecasts. However, in Crochemore et al. (2020) we showed that E-HYPE seasonal forecasts can yield as skilful information as a local model can when looking at anomalies or other statistics relative to model historical time series. In this study, the Lake Como operations were optimized using E-HYPE historical time series so that operations are informed by seasonal forecast anomalies. We will discuss this point in the revised version of the paper.

*Crochemore, L., M.H. Ramos, and I.G. Pechlivanidis (2020), Can Continental Models Convey Useful Seasonal Hydrologic Information at the Catchment Scale?, Water Resources Research, 56(2)*

4. Finally, the manuscript would benefit from considering a section summarizing the limitations of the study and ways to overcome these limitations. This could be included in the discussion section.

We thank the reviewer for the suggestion and we will add such discussion about the limitation of the study, including the continental vs local scale model from the previous point, in the last section of the revised manuscript.

Considering these, and the fact that the scientific significance and quality are excellent, my suggestion to the editors is to accept after minor revision in the context of my specific and technical comments. I am listing a number of suggestions in the form of technical comments that will improve the presentation of the study.

We thank the reviewer for the positive and constructive comments.

TECHNICAL COMMENTS L200: The simulation horizon for the policy optimization is 2007-2015 while results are presented for the 1996-2008 period (thereafter). In case

this is correct, is there any effect from potentially different operation policies between these two periods (considering also that the 2005 drought is out of the 2007-2015 bound)?

This is a typo and all experiments refer to the horizon 1996-2008.

Figure 4: Please consider adding a straight line in panels (b) and (c) indicating flood level.

We thank the reviewer for the suggestion and we will revise the figure accordingly.

L240-241: This is not clear in the figure. Please explain.

The comment refers to the similarity of the baseline, ESP, and SYS4 trajectories, which are on average almost overlapped until the third week of June, while they look more separated during the drawdown period with the SYS4 that is able to keep a high level also in July. We will rephrase this comment in the revised manuscript.

L249-250: but also less efficient onwards (from July to mid-August).

The comment by the reviewer is correct, ESP and SYS4 reach lower levels than the baseline in the second half of the 2005 summer. Yet, this strategy is not necessarily less efficient and can also be considered as an extreme drought mitigation measure triggered by the extreme drought conditions predicted for August in order to support a more reliable irrigation supply than under the baseline by sacrificing few extra centimeters of lake level. We will add a sentence to discuss this aspect in the revised version of the paper.

Figure 5: could consider adding a second panel on the right illustrating the benefit with respect to the baseline.

We thank the reviewer for the suggestion and we will revise the figure accordingly.

Table 2: Based on the values in the table, does the optimum profit comes from informing farmers with the minimum values (SYS4-min)?

Yes, this is correct. The minimum of the ensemble results in the best forecast looking at the performance over the full period. However, for the extreme drought of 2005, the 25th percentile would perform better.

L279: Please provide more information on the behavioral factors.

We model behavioral factors capturing different levels of risk aversion in the interpretation of the uncertainty associated to the forecast ensemble: we first explore decisions that are dependent on the ensemble mean and then move to more and more risk averse behaviors that condition the decision on low percentiles of the ensemble, thus looking at the more negative conditions in terms of irrigation supply. We will better describe and motivate the behavioral factors in the revised version of the paper.

---

## Editor Comment (EC1) · Dimitri Solomatine (Editor) · 20 Apr 2020

The paper received very well prepared reviews, with clear recommendations. The evaluations is overall quite positive. The authors replies show, that these comments and recommndations are understood, (most of them) are accepted, and they present clear plan how the manuscript will be revised. They are invited to do so. I wish the authors good luck in doing this, during these difficult times...

---

## Author Response (AR1)

POLITECNICO
MILANO 1863

Reply to reviewers about paper hess-2019-659

**From skill to value: isolating the influence of end-user behaviour on seasonal forecast assessment**

Matteo Giuliani, Louise Crochemore, Ilias Pechlivanidis, Andrea Castelletti

Matteo Giuliani, Assistant Professor
Department of Electronics, Information, and Bioengineering, Politecnico di Milano

Via Ponzio 34/5, 20133 Milano, Italy
Tel: +39 02 2399 9040
E-mail: matteo.giuliani@polimi.it

Dear Editor,

We would like to thank you and the two anonymous reviewers for the comments and suggestions. In preparing the response to reviewers, we used the following rules: references to line numbers and figures are all to the revised manuscript; authors' replies are in blue; brief text additions are reported in blue italics.

**Reviewer#1**
The subject of the paper "From skill to value: isolating the influence of end-user behaviour on seasonal forecast assessment" is of direct interest to the Journal of Hydrology and Earth System Sciences. Authors introduce and apply a framework in the context of valuing the potential benefit of seasonal forecast in terms of economic end users benefit. Although there are several aspects that need to be further elaborated, this is a step forward in moving from skill to impact (financial) based assessments.

1. One of my concerns is that the examination of the value of forecast is limited to a single lead time (51 days ahead) and the potential benefit of other lead times to the current framework are not examined, or at least discussed.
Our analysis focuses on a specific forecast lead time that was identified in a previous work, i.e., Denaro et al. (2017), as the most valuable for improving Lake Como operations. In that work, we did indeed comparatively analyze forecasts over different lead times from 1 week to 60 days. We clarified this point at lines 206-208 of the revised manuscript.

Denaro, S., D. Anghileri, M. Giuliani, and A. Castelletti (2017), Informing the operations of water reservoirs over multiple temporal scales by direct use of hydro-meteorological data, Advances in Water Resources, 103, 51–63

2. Crop yield modeling is an integral part of the valuing framework. The simulation of crop production is based on water availability and growing degree days controlled by temperature. From the information provided in the manuscript it is not clear whether the heat unit module of agricultural model is also informed by seasonal forecasts.
The agricultural model is not informed by the forecasts because our analysis investigates the value of forecasts in informing the Lake Como operation that provides the irrigation supply to the agricultural districts considering as water demand the sum of the water rights of the different users, which therefore does not vary across years. Conversely, we are not exploring here decisions by the farmers that could benefit from the seasonal forecasts, but we studied this problem in a previous work (see Li et al., 2017). In the revised manuscript, we clarified the

decisions we are considering as well as the definition of the irrigation demand at lines 188-191.

Li, Y., M. Giuliani, and A. Castelletti (2017), A coupled human–natural system to assess the operational value of weather and climate services for agriculture, Hydrology and Earth System Sciences, 21, 4693-4709

3. I understand that the present study, beside other components, examines the usefulness/applicability of a continental scale hydrological model (E-HYPE) with known issues in simulating streamflow dynamics due to local scale hydrological features (as referred in L120-125 – constant positive bias of E-HYPE / failure in seasonal dynamics).The question is whether the use of fine-tuned local scale model would increase the performance of the overall system?

We agree with the reviewer that a fine-tuned local scale model may in principle increase both the skill and the value of the forecasts. However, in Crochemore et al. (2020) we showed that E-HYPE seasonal forecasts can yield as skillful information as a local model can when looking at anomalies or other statistics relative to model historical time series. In this study, the Lake Como operations were optimized using E-HYPE historical time series so that operations are informed by seasonal forecast anomalies. We added a discussion about this point in the revised version of the paper (see lines 144-148 and 380-382).

Crochemore, L., M.H. Ramos, and I.G. Pechlivanidis (2020), Can Continental Models Convey Useful Seasonal Hydrologic Information at the Catchment Scale?, Water Resources Research, 56(2)

4. Finally, the manuscript would benefit from considering a section summarizing the limitations of the study and ways to overcome these limitations. This could be included in the discussion section.

We thank the reviewer for the suggestion and we added such discussion about the limitation of the study, including the continental vs local scale model from the previous point, in section 4.4 of the revised manuscript (see lines 374-386).

Considering these, and the fact that the scientific significance and quality are excellent, my suggestion to the editors is to accept after minor revision in the context of my specific and technical comments. I am listing a number of suggestions in the form of technical comments that will improve the presentation of the study.

We thank the reviewer for the positive and constructive comments.

TECHNICAL COMMENTS
L200: The simulation horizon for the policy optimization is 2007-2015 while results are presented for the 1996-2008 period (thereafter). In case this is correct, is there any effect from potentially different operation policies between these two periods (considering also that the 2005 drought is out of the 2007-2015 bound)?

This is a typo and all experiments refer to the horizon 1996-2008.

Figure 4: Please consider adding a straight line in panels (b) and (c) indicating flood level.

We thank the reviewer for the suggestion and we revised the figure accordingly.

L240-241: This is not clear in the figure. Please explain.

The comment refers to the similarity of the baseline, ESP, and SYS4 trajectories, which on average almost overlap until the third week of June, while they look more separated during the drawdown period, with SYS4 being able to keep a high level also in July. We rephrased this comment in the revised manuscript (see lines 277-280).

L249-250: but also less efficient onwards (from July to mid-August).

The comment by the reviewer is correct, ESP and SYS4 reach lower levels than the baseline in the second half of the 2005 summer. Yet, this strategy is not necessarily less efficient and can also be considered as an extreme drought mitigation measure triggered by the extreme drought conditions predicted for August in order to support a more reliable irrigation supply than under the baseline by sacrificing few extra centimeters of lake level. We added a comment to discuss this aspect in the revised version of the paper (see lines 289-293).

Figure 5: could consider adding a second panel on the right illustrating the benefit with respect to the baseline.

We thank the reviewer for the suggestion and we revised the figure accordingly.

Table 2: Based on the values in the table, does the optimum profit comes from informing farmers with the minimum values (SYS4-min)?

Yes, this is correct. The minimum of the ensemble results in the best forecast looking at the performance over the full period. However, for the extreme drought of 2005, the 25th percentile would perform better. Note that Table 2 is now Table 3 is the revised manuscript.

L279: Please provide more information on the behavioral factors.

We model behavioral factors capturing different levels of risk aversion in the interpretation of the uncertainty associated to the forecast ensemble: we first explore decisions that are dependent on the ensemble mean and then move to more and more risk averse behaviors that condition the decision on low percentiles of the ensemble, thus looking at the more negative conditions in terms of irrigation supply. We better described and motivated the behavioral factors in the revised version of the paper (see lines 65-81).

Reviewer #2

This is a study into the value of inflow forecasts in water release decision making, focusing on the benefits to agricultural profitability. Previous studies have demonstrated how forecasts can be adopted in reservoir operations to marginally improve on a prespecified objective. This paper offers an incremental advance by coupling the reservoir model to an agricultural model, allowing for calculation of profits associated with the updated release schedule. The subject is certainly of general interest to the hydrological community. The paper is well written and the method easy to follow. While the study is mostly sound (I have a few concerns outlined below), a significant contribution to knowledge is missing. One can easily deduce without this study that reduced water supply deficits in a reservoir release model should lead to increased profits for crop-growers relying on that water. The monetary values provided cannot be offered as a contribution either, since they are not reflective of actual profit gains that would be gleaned by crop-growers (partly because the reservoir operations are stylized for this case study rather than representing real world operations). The approach taken is described as a "novel evaluation framework". It appears to be a forecast product providing information for a reservoir model, the release from which forces an agricultural profit model. One-way coupling of models (which is what I understand this to be) does not qualify as a novel framework in my view. Lastly, because the study is conducted on a single site and using a short time series with only one drought (with much of the analysis drawn from performances during that drought), the conclusions are not generalizable. The authors acknowledge this lack of generalizability in their final sentences, and I think that their suggestions for future research are actually required in this paper to help with the knowledge contribution. While a single case study can be valuable, I cannot see compelling new insights on value of forecasts arising from this analysis to warrant publication. I think the suggested exponential relationship between forecast skill and farmer profit could be a significant contribution if demonstrated and elaborated more carefully through more detailed analysis across a range of possible droughts and with incremental adjustments to the forecast skill. I would be supportive of publication of this paper if the authors can (a) deepen their analysis to generate a more compelling advance on existing knowledge, and (b) address the small number of other concerns listed below.

We agree with the reviewer that our evaluation framework per se may not represent a sufficient contribution to the existing literature. However, in our opinion there are aspects other than the integrated modelling chain that are novel, such as the use of different river flow forecasts as inputs to understand which part of the hydrological modelling chain is relevant in this case, as well as the inclusion of the decision maker's perspective by looking at specific forecast quantiles. In the revised version of the paper, we clarified that these two aspects, along with the inference of a relation between gains in forecast skill and gains in end-user profit, represent the main methodological contributions of the paper (see lines 51-81).

Moreover, we respectfully disagree that our quantification of the value of hydroclimatic services in terms of estimates of potential economic benefit to the endusers can be summarized as a "forecast product providing information for a reservoir model, the release from which forces an agricultural profit model". Our evaluation framework combines a state-of-the-art hydroclimatic service run by SMHI with a detailed model of the Lake Como basin. This latter couples an advanced operational module to simulate the lake operation, including an optimization of the operational decisions via Reinforcement Learning techniques, with an accurate description of the agricultural district provided by a spatially distributed model with a regular mesh of cells with a side length of 250 m. Previous works (e.g., Giuliani et al., 2016) demonstrated that this model is capturing the main characteristics of the real systems, including the actual operations, and its outputs were validated with respect to observed data both in terms of lake releases and of agricultural practices such as water requirements and consumption, crop production, or land use decisions. We therefore consider our estimates to be a valuable contribution for the case study area and in the revised version of the paper, we better described the potential value of our results for the considered case study (see lines 61-64).

Specific comments:
The decision to use quadratic water supply deficit as the objective function is not fully justified. If the purpose of the water supply is to meet farmer needs, and if profit is the goal of the farming community, then why not use farmer profit as the objective? This would greatly improve the interpretability of the results, particularly for aim (iii) "the inference of the relation between gains in forecast skill and gains in end-user profit." Currently, the paper lacks discussion on how the discontinuity between the optimization objective and profitability affects the conclusions drawn. In particular, the squaring of the water deficit objective would normally result in hedging behavior that would reduce overall profit to avoid possible cases of extreme deficit. It's also not clear from this analysis how water deficit affects profit. Does a small deficit necessarily imply loss of crop production, or can farms run at full profitability during periods of small or moderate deficit?

The quadratic water supply deficit is a traditional formulation in reservoir operations since the work by Hashimoto et al. (1982). This objective generates hedging strategies that minimize large deficits, while accepting small, distributed deficits; these strategies are known to be suitable for irrigation practices as crops can resist in case of small deficits while extreme ones can generate crop failures. Obviously, the larger the deficit, the larger the difference between potential and actual evapotranspiration computed in the crop-yield module of the agricultural district, with this delta generating large stress and loss of production according to the approach proposed in FAO (Doorenbos et al., 1979) and implemented in our model. For example, moving along the baseline Pareto front from the policy selected in Figure 4 towards solutions that attain lower deficits (e.g., a policy P1 with squared deficit equal to 2749 $(m^3/s)^2$ or policy P2 with squared deficit equal to 2672 $(m^3/s)^2$) generates higher profits for the farmers, specifically 24.6 M€/year for P1 and 24.7 M€/year for P2. Moreover, the computation of the water supply deficit includes a time-varying parameter that penalizes more the deficit after germination to the

beginning of phenological maturity, with these crop stages determined by the agricultural district model. We clarified this aspect in the revised version of the paper (see lines 110-117).

The reason for not directly using the farmer profit as an objective function relies in the computational requirements of the agricultural model simulations (which is based on a mesh of about 11,000 cells with a side length of 250 m) that are incompatible with the computational costs of the EMODPS approach used for the design of the optimal Lake Como operations. The EMODPS optimization indeed requires running 40 million simulations for each forecast input, and the overall analysis comprises a total of 320 million simulations that required approximately 42,670 computing hours. We referred to this point in the revised version of the paper (see lines 235-238).

Lastly, it is worth mentioning that the validation of the model in Giuliani et al. (2016) showed that a policy designed using this formulation generates a good approximation of the observed operations of the lake.

Doorenbos, J., Kassam, A., and Bentvelsen, C.: Yield response to water, irrigation and drainage. Paper no. 33, Tech. Rep., Food and Agriculture Organization, Rome, Italy, 1979.

The decision to vary the ensemble selected from mean to 10th and 25th percentiles to capture drought risk aversion requires better justification, too. It would seem more prudent to adjust the objective to represent risk-averse preference (e.g., increasing the exponent applied to the objective, or, if changing the objective function to farmer profit, adding penalties for very significant losses) than to deliberately under-estimate the inflow.

The idea of exploring alternative interpretations of the forecast ensemble by replacing the mean with low percentiles is motivated by the growing literature suggesting that, at seasonal time scales, probabilistic forecasts are often used to convey uncertainties related to initial hydro-climatic conditions, scenarios of predicted meteorological conditions, and adopted models, potentially adding value for decision making (see Georgakakos and Graham, 2008; Cloke and Pappenberger, 2009). At the same time, there is also growing evidence that higher forecast accuracy does not necessarily imply better decisions because of the challenges associated to the human interpretation of forecasts as well as to the communication of probabilistic information (Ramos et al., 2010, 2013; Crochemore et al., 2016). However, at the best of our knowledge, this point has been so far investigated mostly via serious games, interviews, or direct interactions with decision makers, while our work aims at providing a quantitative analysis of this challenge by simulating how different behavioral attitudes influence the interpretation of the forecast ensemble and ultimately impact on operational decisions and resulting performance. We better clarified this contribution in the revised version of the paper (see lines 65-81).

Cloke, H. and Pappenberger, F. (2009), Ensemble flood forecasting: a review, Journal of Hydrology, 375, 613–626

Crochemore, L., Ramos, M., Pappenberger, F., van Andel, S., and Wood, A. (2016), An experiment on risk-based decision-making in water management using monthly probabilistic forecasts, Bulletin of the American Meteorological Society, 97, 541–551

Georgakakos, K. and Graham, N. (2008), Potential benefits of seasonal inflow prediction uncertainty for reservoir release decisions, Journal of Applied Meteorology and Climatology, 47, 1297–1321

Ramos, M., Mathevet, T., Thielen, J., and Pappenberger, F. (2010), Communicating uncertainty in hydrometeorological forecasts: mission impossible?, Meteorological Applications, 17, 223–235

Ramos, M., van Andel, S., and Pappenberger, F. (2013), Do probabilistic forecasts lead to better decisions?, Hydrology and Earth System Sciences, 17, 2219–2232, doi:10.5194/hess-17-2219-2013

Line 81: please clarify what "heavily man-overworked" means (and why its relevant).
We meant that the water resources in the basin are highly regulated by water infrastructures, including 16 Alpine hydropower reservoirs in the upstream part of the catchment that can store about 545 $Mm^3$, which is more than twice the active volume of the lake; Lake Como in the middle, which is a deep glacial lake whose outlet is controlled since 1946 with the twofold primary purpose of water allocation to the downstream users and flood protection along the lake's shoreline, along with additional interests related to navigation, fishing, tourism, ecosystems; the lake release serves a dense network of downstream irrigation canals, which convey water to four agricultural districts with a total surface of 1400 $km^2$, mostly growing maize. The same releases are also sufficient to feed eight run-of-river hydroelectric power plants. These features are peculiar characteristics of this system, which should not be confused with a natural lake, and the resulting high level of control of the water in the basin is an important factor that motivates the search for more efficient management strategies relying on hydroclimatic services. We rephrased this sentence in the revised version of the paper (lines 86 and 100-103).

Line 89: do you mean "most Southerly" point on the lake shoreline, or the "near the outflow" of the lake?
We mean lowest point in terms of elevation, which is the reason why it is the location suffering the most from the floods. Note that the lake outflow are in the other branch of the lake, while the one where Lake Como is located is a dead branch. We rephrased the original sentence to clarify this point (see lines 97-98).

Line 257: Why bother with the Pareto analysis if the flood objective effectively becomes a constraint. I don't think the readers of the study need all of the detail of the Pareto analysis if multi-objective optimization is not actually used to generate the key results.

The flood objective is not a constraint in our problem, but we designed the Pareto optimal set of operating policies by using a truly multi-objective approach, namely the Evolutionary Multi-Objective Direct Policy Search (Giuliani et al., 2016). Since the result is then a set of solutions that explores the tradeoff between flood control and water supply, we used a reference value of flood days only to filter the Pareto optimal solutions and select one policy for each set, attributing their different performance to the different forecasts that they use. However, it is important to notice that the benefit of informing the lake operations with hydroclimatic services is in both the objectives, with the overall Pareto front that shifts toward the bottom-left corner of the objective space. We better clarified this point in the revised manuscript by reinforcing the tradeoff analysis narrative prior to focusing on the selected policies, specifically by reporting the values of hypervolume indicator to quantify the improvement of the full Pareto optimal set (see lines 171-176; 241-243; 257-262 and the newly introduced Table2).

Giuliani, M., Castelletti, A., Pianosi, F., Mason, E., and Reed, P. (2016), Curses, tradeoffs, and scalable management: advancing evolutionary multi-objective direct policy search to improve water reservoir operations, Journal of Water Resources Planning and Management, 142

Line 260: the fact that profits are improved through operations is used to support the idea that forecasts can be valuable for managing extreme drought. Presumably the impact is greatest during drought because this is the only time when profit can be compromised (i.e., average flow conditions are unlikely to lead to supply deficits, meaning forecasts are not actually useful except leading up to and during drought). Is this correct? If so, why not focus analysis on droughts and also introduce other drought events to help support and generalize these conclusions?
In the case of Lake Como, the role of operations is larger than what the reviewer says because the natural water availability (i.e., the lake inflows) is not covering the downstream demand and the system would experience deficits during the summer. This is the reason why hydroclimatic services are expected to be valuable also in normal conditions, and likely also in wet years as they would suggest the operator to keep a larger flood buffer by releasing more water than in normal conditions as high inflow volumes are expected over the upcoming months. We clarified the central role of the lake operations in the revised version of the paper (see lines 91-94).

Line 311: Has this function been fitted across all of the points on Figure 6? Please justify or comment on the appropriateness of combining the all-years and 2005 results in the same function. The idea of exponential relationship between profit and forecast skill would be a powerful conclusion, but is surely best demonstrated using (a) a model that can adjust forecast skill incrementally allowing generation of many data points, and (b) repeating the analysis across multiple droughts.
Yes, the function is fitted across all the points in the Figure. We agree with the reviewer that having more points would make this result more statistically sound.

However, as mentioned also in previous replies, the data/modeling/computational requirements of our analysis are not negligible, thus limiting the possibility of easily generating more points. We therefore consider this result anyway acceptable in the context of our paper, where this function is one out of three contributions, and we discussed the associated limitations of such analysis in the revised version of the manuscript (see lines 374-379).

[revised manuscript text omitted]

---

## Author Response (AR2)

**POLITECNICO**
MILANO 1863

Reply to reviewers about paper hess-2019-659

**From skill to value: isolating the influence of end-user behaviour on seasonal forecast assessment**

Matteo Giuliani, Louise Crochemore, Ilias Pechlivanidis, Andrea Castelletti

Matteo Giuliani, Assistant Professor
Department of Electronics, Information, and Bioengineering, Politecnico di Milano

Via Ponzio 34/5, 20133 Milano, Italy
Tel: +39 02 2399 9040
E-mail: matteo.giuliani@polimi.it

Dear Editor,

We would like to thank you and the two anonymous reviewers for the comments and suggestions. In preparing the response to reviewers, we used the following rules: references to line numbers and figures are all to the revised manuscript; authors' replies are in blue; brief text additions are reported in blue italics.

Editor

Dear authors: finally I have both reviews. Reviewer 2 recommends the paper for publication, without poviding comments, putting all judgments at 'excellent'. Reviewer 1 however, recommends 'major revision', provides extensive comments, and asks for more attention to the previsously made comments (at Discussion phase), and for deeper revisions, and perhaps certain reformulations of generalisations ("claims" as the reviewer puts it). Please address them since I think taking them into account would contribute to the the quality of the paper - or present a rebuttal if you don't agree with them.

Good luck!

We thank the editor for the encouraging and constructive comments. As suggested, we improved the manuscript by working on the following main aspects:

- We better clarified that the paper contribution is not the evaluation framework itself, but rather the analyses we conducted using this framework as our methodology, namely (1) the isolation of the part of the hydrological modelling chain mostly contributing to the estimated forecast value, (2) the assessment of the sensitivity of the results on different end-user interpretations of the probabilistic forecast information, (3) the inference of a relation between gains in forecast skill and end-user profit.
- We reinforced the inference of skill-value relationship by using annual results that allowed increasing the number of samples used for the regression to 65 (see new Figure 6 in the revised manuscript).
- We added the complete formulation of forecast skill.

**Reviewer#1**

No comments, suggested "accepted as is".

We thank the reviewer for the positive evaluation of the paper.

**Reviewer#2**

Rather than rework their study in response to comments raised, the authors have opted to provide a full rebuttal. I'm afraid I remain unconvinced of the key contribution having carefully considered the responses.

We respectfully disagree with the reviewer comment. We provided 10 pages of detailed replies (6 dedicated to R#2 comments), leading to major edits of the paper (i.e. 95 lines of track changes, that correspond to about 23% of the paper). Far from being a full rebuttal, our reply once again tried to better clarify the contribution of the paper following the reviewer indications.

In brief:

- The authors retain their claim that the paper provides a novel framework, based on the argument that the input data are state-of-the-art, reservoir model is accurate, and so on. I don't doubt any of these claims. However, for the framework to be novel it must involve more than one-way coupling of forecasts -> reservoir model -> ag profit model. This is just linking output of one model to another, and not a novel framework.

In revising the manuscript the first time, we actually tried to clarify that the framework itself is not the novel contribution of our paper (we removed all the occurrence of "novel framework"). Following the reviewer suggestion, we further adjusted our text to better stress that the framework is part of our methodology, while the novel contributions of the work are the type of analyses we conducted, namely (1) the isolation of the part of the hydrological modelling chain mostly contributing to the estimated forecast value, (2) the assessment of the sensitivity of the results on different end-user interpretations of the probabilistic forecast information, (3) the inference of a relation between gains in forecast skill and end-user profit.

- I agree with the authors that the value Euro490million is a valuable insight, particularly for those with an interest in the region. But it's very case specific and not a broad enough contribution to knowledge for a scientific paper.

Following the reviewer suggestion, a section discussing the limitations and future research, including this point, was already added during the previous round of review and we are fully aware that our results might not be fully generalizable. However, we argue that the selection of our case study located in the region with the highest share of irrigated areas in Europe (Eurostat 2019) makes our findings relevant for a scientific publication.

In addition, there is a general growing interest in studying real systems by developing tailored applications to the specific needs and requirements of the considered case study (see, for example, Nayak et al. (2018), Neumann et al.

(2018), Delorit and Block (2019), Anghileri et al. (2019), Alexander et al. (2020), Macian-Sorribes et al. (2020)).

o   Alexander et al. (2020), Forecast-informed reservoir operations to guide hydropower and agriculture allocations in the Blue Nile Basin, Ethiopia, International Journal of Water Resources Development
o   Anghileri et al. (2019). The value of subseasonal hydrometeorological forecasts to hydropower operations: How much does preprocessing matter? Water Resources Research, 55, 10159–10178.
o   Delorit, J. and Block, P. (2019), Using Seasonal Forecasts to Inform Water Market-Scale Option Contracts, Journal of Water Resources Planning and Management, 145
o   Eurostat (2019), Agri-environmental indicator - irrigation, https://ec.europa.eu/eurostat/statistics-explained/index.php/ Agri-environmental_indicator_-_irrigation#Analysis_at_regional_level.
o   Macian-Sorribes et al. (2020). Fuzzy post-processing to advance the quality of continental seasonal hydrological forecasts for river basin management. Journal of Hydrometeorology, 1-47.
o   Nayak, M., Herman, J., and Steinschneider, S. (2018), Balancing Flood Risk and Water Supply in California: Policy Search Integrating Short-Term Forecast Ensembles With Conjunctive Use, Water Resources Research, 54, 7557–7576
o   Neumann et al. (2018). The 2013/14 Thames Basin floods: Do improved meteorological forecasts lead to more skillful hydrological forecasts at seasonal time scales?. Journal of Hydrometeorology, 19(6), 1059-1075.

- Modeling risk aversion of decision makers through use of different forecast percentiles is still poorly justified in my view.

This point represents one of the main contributions of the paper as the assessment of forecast value is filtered by the way end-users make use of the provided information. In the literature, there is growing evidence that higher forecast accuracy does not necessarily imply better decisions because of the challenges associated to the human interpretation of forecasts as well as to the communication of probabilistic information (Ramos et al., 2010, 2013; Crochermore et al.,2016).

Individual behaviours and risk perceptions – which can be modeled by using different metrics/statistics to filter the uncertainty in forecast information (see, for example, Peterson 2017) - therefore play a key role in influencing the end-user assessment of probabilistic seasonal forecast value. So far, this point has been investigated mostly via serious games, interviews, or direct interactions with decision makers, while our work aims at providing a quantitative analysis of this challenge by simulating how different behavioral attitudes (modeled by specific forecast quantiles capturing increasing levels of drought risk aversion) influence the interpretation of the forecast ensemble and ultimately impact on operational decisions and resulting performance.

o   Crochemore et al. (2016), An experiment on risk-based decision-making in water management using monthly probabilistic forecasts, Bulletin of the American Meteorological Society, 97, 541–551
o   Peterson, M. (2017). *An introduction to decision theory*. Cambridge University Press (second edition).
o   Ramos et al. (2010), Communicating uncertainty in hydro- meteorological forecasts: mission impossible?, Meteorological Applications, 17, 223–235

o   Ramos et al. (2013), Do probabilistic forecasts lead to better decisions?, Hydrology and Earth System Sciences, 17, 2219–2232

- The authors continue to press the claim of an exponential relationship given the very limited nature of the inputs and the fact that full simulation period results are combined with 2005-only results in this graphic. The authors acknowledge that this is problematic, conceding that more points would make the analysis more statistically sound. The authors' response is that generating more points is too computationally intensive. If this is the case, then the associated conclusion needs to be removed or properly caveated. The abstract reads "…our results show an exponential skill to value relation where large gains in forecast skills are necessary to generate moderate gains in end-user profit". This is a powerful statement and must be backed up with rigor.

In the revised version of the paper, we have now improved our analysis by inferring the skill-value relation using more data samples. The new figure 6 relies on 65 points, representing the individual values of skill and profit for all the 13 years simulated in our study. Moreover, we also investigate how the relation is impacted by hydrologic conditions as well as by the isolated source of forecast value (see the new Figure 6 in the revised manuscript, also reported below). While still being not fully statistically rigorous, we believe these new results are now more robust as they cover a wider range of hydroclimatic conditions, that are also impacting on the inferred relation.

[Figure]

*Figure 6: Scatterplot between forecast skill and value. The color of the circles represents the hydrologic conditions (i.e., annual inflow to the lake) of the different years. Markers represent either an isolated component of the forecast system or a behavioral factor for each year over the 1996-2008 time period: all components (panel a), ESP vs baseline (panel b), SYS4-mean vs ESP (panel c), SYS4-min vs SYS4-mean (panel d), SYS4-p10 vs SYS4-mean (panel e), SYS4-p25 vs SYS4-mean (panel f).*

(Also, I don't think "skill" is actually defined qualitatively or mathematically anywhere in the paper)

The skill is defined as a function of the ratio in Root Mean Square Error between forecast systems and benchmarks from Table 3. We added this formulation as reported below in the revised version of the paper (see lines 350-353):

[revised manuscript text omitted]